# Determinants of soil-transmitted helminth infections among pre-school-aged children in Gamo Gofa zone, Southern Ethiopia: A case-control study

Mekuria Asnakew Asfaw[1]*, Teklu Wegayehu[2], Tigist Gezmu[1], Alemayehu Bekele[1], Zeleke Hailemariam[3], Teshome Gebre[4]

1 Collaborative Research and Training Centre for NTDs, Arba Minch University, Arba Minch, Ethiopia,
2 Department of Biology, College of Natural Sciences, Arba Minch University, Arba Minch, Ethiopia, 3 School of Public Health, College of Medicine and Health Sciences, Arba Minch University, Arba Minch, Ethiopia,
4 The Task Force for Global Health, International Trachoma Initiative, Addis Ababa, Ethiopia

* maksambaramr23@gmail.com

**Data Availability Statement:** All relevant data are within the manuscript and its Supporting information files.

## Abstract

### Background

Pre-school aged children (PSAC) are highly affected by soil-transmitted helminths (STH), particularly in areas where water, sanitation, and hygiene (WASH) are inadequate. Context-specific evidence on determinants of STH infections in PSAC has not been well established in the study area. This study, therefore, aimed to fill these gaps in Gamo Gofa zone, Southern Ethiopia.

### Methods

A community-based unmatched case-control study, nested in a cross-sectional survey, was conducted in January 2019. Cases and controls were identified based on any STH infection status using the Kato-Katz technique in stool sample examination. Data on social, demographic, economic, behavioral, and WASH related variables were collected from primary caregivers of children using pre-tested questionnaire. Determinants of STH infections were identified using multivariable logistic regression model using SPSS version 25.

### Results

A total of 1206 PSAC (402 cases and 804 controls) participated in this study. Our study showed that the odds of STH infection were lowest among PSAC living in urban areas (AOR = 0.55, 95% CI: 0.39–0.79), among those from households with safe water source (AOR = 0.67, 95% CI: 0.47–0.0.93), and in those PSAC from households with shorter distance from water source (<30 minutes) (AOR = 0.51, 95% CI: 0.39–0.67). On the other hand, the odds of STH infection were highest among PSAC from households that had no functional hand washing facility (AOR = 1.36, 95% CI: 1.04–1.77), in those PSAC from households that had unclean latrine (AOR: 1.82, 95% CI: 1.19–2.78), and among those PSAC under caregivers

**Funding:** This study is made possible by the generous support of the Collaborative Research and Training Center, Arba Minch University, Ethiopia.

**Competing interests:** The authors have declared that no competing interests exist.

who had lower score (≤5) on knowledge related to STH transmission (AOR = 1.85, 95% CI: 1.13–3.01).

## Conclusions

Given efforts required eliminating STH by 2030; the existing preventive chemotherapy intervention should be substantially strengthened with WASH and behavioral interventions. Thus, an urgent call for action is required to integrate context-specific interventions, particularly in rural areas.

## Introduction

According to the World Health Organization (WHO) estimate, soil-transmitted helminths (STH), including *Ascaris lumbricoides, trichuris trichiura, and* hookworms, affect more than 2 billion people worldwide [1]. It is known that pre-school aged children (PSAC) (1–5 years) account for significant proportion (10%-20%) of the people affected with STH [2, 3]. STH infections among children have adverse health outcomes, such as anaemia, malnutrition [4], stunting [5], and cognitive impairment [6]. In Ethiopia, STH are among the most prevalent Neglected Tropical Diseases (NTDs), with about 81 million people living in STH endemic areas, of which 9.1 million are PSAC [7]. Infections with STH are primarily linked with poverty, with the highest prevalence rates found in developing countries, where hygiene and sanitation are absent or inadequate, and access to safe, clean water is insufficient and inaccessible [6, 8–10].

Preventive chemotherapy, deworming, using annual or biannual single-dose albendazole (400 mg) or mebendazole (500 mg) is recommended by the WHO as a public health intervention against STH [11]. It has been provided for PSAC as one of the high risk group for many years in endemic countries including Ethiopia in areas where the baseline prevalence of any STH infection is 20% or higher among children in order to control and eliminate STH [1, 7, 11]. Since 2005, Ethiopia has been applying mass drug administration (MDA) against STH to a large number of PSAC, with coverage of 78% in 2009 [1]. However, treatment only does not halt the cycle of transmission. Evidences suggest that improvements of Water, Sanitation and Hygiene (WASH) infrastructures and appropriate health seeking behavior are indispensable to achieve sustained control and elimination of STH [12, 13]. Fortunately, a recent development of the WASH–NTD joint strategy provides an entrance point and guidance for improved communication, coordination, and collaboration [14].

To sustain the achievements made by deworming activities, and eliminate STH by 2030 (<2% proportion of STH infections of moderate and heavy intensities), previous study and WHO's road map recommend context-specific WASH interventions [15, 16]; this is due to the fact that the association between WASH and STH infection is complex. Although there is an increased emphasis on the role of WASH on STH control, evidence gaps still exist in our understanding of the association between WASH and STH infection in Ethiopia—there is paucity of evidences regarding context-specific risk factors associated with STH infections in pre-school aged children. In addition, previous studies conducted in Ethiopia are cross-sectional, which had limitation in terms of identifying determinants of STH infections. This study, therefore, aimed to identify determinants of STH infections in PSAC in Gamo Gofa zone, Southern Ethiopia.

## Methods

### Study setting

This study was conducted in five districts of the former Gamo Gofa zone, in the Southern Nations, Nationalities, and Peoples' Regional State of Ethiopia. The zone had 15 districts and two city administrations. A total of 2,043,668 people (1, 013,533 males and 1,030,135 females) live in the zone, according to the 2007 census and projections of Central Statistical Agency of Ethiopia [17]. It is known that STH is endemic in the zone [7].

### Study design and period

A community based unmatched case-control study, which was nested in a community based cross-sectional survey, was conducted in January 2019. First, cases (a group known to have STH infection) and controls (a group known to be free of the STH infection) were identified, and then traced back to investigate exposures to potential risk factors.

### Source and study populations

The source population was all PSAC in Gamo Gofa zone, and the study population (cases and controls) was all selected PSAC in the selected STH endemic *kebeles* (localities). Since there is no consistent definition for PSAC in current literatures, all children aged 1 to 5 years who are not yet attending primary school were considered as pre-school aged children, as supported by WHO guideline [18].

### Inclusion and exclusion criteria

Cases and controls were selected irrespective of infection intensity. Some eligible children were excluded in the event when caregivers were unavailable to provide their information.

### Sample size estimation

The sample size was determined using double proportion formula, using Open Epi version 2.3.1, by considering the following into considerations: 80% power; ratio of controls to cases (2:1); two-sided confidence level (1-α); 95% confidence interval; prevalence of exposure among cases (6.16%), and prevalence of exposure among controls (2.58) [19]; hence we estimated a sample size of 402 cases and 804 controls.

### Sampling strategy

Both cases and controls were systematically selected from 5 districts of the zone by taking probability proportional-to-population size into account based on the number of cases in the cross-sectional study [20]; which was conducted ahead of this study in the same study area (Table 1).

### Study variables

In this study, STH infection status (positive or negative for any STH) was the outcome variable, and the independent variables were socio-demographic and economic factors; child factors; receiving deworming treatment in the last year; WASH factors, and knowledge and practice (KP) of caregivers on transmission and prevention of STH.

**Table 1. Sampling technique.**

| Category | Districts | | | | | Total sample size |
|---|---|---|---|---|---|---|
| | **Deremalo** | **Chencha** | **Dita** | **Demba Gofa** | **Bonke** | |
| **Cases** | 61 | 145 | 76 | 30 | 90 | **402** |
| **Controls** | 122 | 292 | 152 | 59 | 179 | **804** |
| **Total** | 183 | 437 | 228 | 89 | 269 | **1206** |

## Data collection and Kato-Katz technique

**Data and stool collection.** Data on risk factors were collected by trained health professionals using standardized and pre-tested paper based questionnaire through face-to-face interviews. Stool samples were examined using the Kato-Katz technique to determine infection status. The stool samples were collected using clean, leak proof and screw cup container, and transported to nearby health facility using an ice-boxes with frozen ice-packs. The specimens were processed within two hours of receipt or kept in an ice-box where travel time exceeded two hours.

**Kato-Katz technique.** The Kato–Katz technique is the diagnostic method recommended by WHO for monitoring large-scale treatment programmes implemented for the control of STH infections. It was performed as follow: A small amount of stool sample was pressed through a sieve to remove large particles. Part of the sieved stool was then transferred to the hole of a template on a slide using flat-sided spatula. The hole was filled; the template was removed; and the remaining sieved sample was covered with cellophane which had been pre-soaked in glycerol. Then, the microscope slides were inverted and the fecal samples were firmly pressed against the hydrophilic cellophane strip on another microscope slide or on a smooth hard surface. The fecal material was spread evenly between the microscope slide and the cellophane strip; it should be possible to read newspaper print through the smear after clarification. The slide was carefully removed by gently sliding it sideways to avoid separating the cellophane strip or lifting it off. Then, the slide was placed on the bench with the cellophane upwards, and water evaporates while glycerol clears the smear. Finally, the smears were examined in a systematic manner and the number of eggs of each species reported. Later multiply by 24 (for a 41.7 mg template) to give the number of eggs per gram of stool [21].

**Data quality control.** Data quality was ensured by standard operational procedure and close monitoring of data collection process by supervisors. In addition, two slides were prepared for each stool sample in order to increase positive predictive value, and bench aids (pictures of parasites eggs) were displayed on wall, in front of microscopy examination for the purpose of internal reference.

**Data analysis and measurement.** A sample size of 1206 participants (402 cases and 804 controls) included to provide 80% power at P <0.05 to detect risk of any STH infections. First, data were edited, coded and entered into EpiData 4.4.2, and then exported to SPSS software (IBM, version 25) for analysis. Second, goodness of model fitness, interaction effect, multicollinearity (correlation coefficient <0.90), and assumption of Chi-Square test were checked before fitting into multivariable model.

Household's wealth status was computed using principal component analysis, and quintiles of wealth index were created to observe the presence of association with STH infection status. Score out of 11/12 variables' response was computed to determine knowledge and practice of caregivers on STH transmission and prevention by counting value within a case (1 = Yes and 0 = No). In this study, latrine cleanliness was stated as absence of faecal material or any dirt on the upper surface/floor of the latrine, and unsafe water was defined as untreated water

obtained from well, river and spring, whereas safe water defined as water obtained from private or public tap water.

Finally, all potential variables with P ≤0.25 with the outcome variable were entered into multivariable logistic regression model using backward stepwise method to identify determinants of STH infections. P-value <0.05 was considered as statistically significant, and odds ratio at 95% confidence interval was indicated as the precision and strength of association.

### Ethics statement

The study was reviewed and approved by Institutional Research Ethics Review Board of Arba Minch University (reference number: CMHS/11222/111). Oral and written consents were received from district administrators and head of households. Assent was not obtained from PSAC since we believe that caregivers are responsible on behalf of them. Children tested positive for STH were treated with albendazole or mebendazole by health professionals at the end of the study.

## Results

### Socio-demographic and economic characteristics

Nearly 45% (181/402) of cases and 49.3% (396/804) of controls were females; 33.8% (136/402) of cases and 32.1% (258/804) of controls were ≤2 years, and 42.8% (172/402) of cases and 46.4% (373/804) of controls their caregivers did not read and write (Table 2). Details on socio-demographic and economic characteristics are presented in Table 2.

### Infection status by STH species

A total of 804 controls and 402 cases participated in this study. With regard to infection status by each individual STH species, overall, ascariasis was the most prevalent (27.7%), followed by trichiurasis (11.9%) and hookworms (4.6%), and 8% PSAC were infected with two STH species (ascariasis and trichiurasis).

### Univariable and multivariable analyses of factors related to STH infections

During univariable analysis, 16 variables were identified with p-value ≤0.25 in relation to STH infection status, such as place of residence, children's age, age of caregiver, source of water, treat water, distance from water source, latrine cleanliness, faeces or any dirt observed on latrine floor, having functional hand washing facility, washing fruits or vegetables before eating, habit of washing hand after cleaning child, child hand washing habit before meal, child hand washing habit after defecation, attending nursery school (started education), caregiver's mean score on knowledge of STH transmission and mean score on knowledge and practice of prevention of STH.

The variables with P ≤0.25 in univariable logistic regression model were entered into multivariable logistic regression model using the backward stepwise method. The reason for using p ≤0.25 was to improve the chances of remaining potential variables in the multivariable model. After adjusting for potential confounders, the model identified the following variables as determinants of STH infection among PSAC. The odds of STH infection were lowest among PSAC living in urban areas (AOR = 0.55, 95% CI: 0.39–0.79), among those from households with safe water source (AOR = 0.67, 95% CI: 0.47–0.0.93), and in those PSAC from households with shorter distance from water source (<30 minutes) (AOR = 0.51, 95% CI: 0.39–0.67). On the other hand, the odds of STH infection were highest among PSAC from households that had no functional hand washing facility (AOR = 1.36, 95% CI: 1.04–1.77), in those PSAC from

**Table 2. Socio-demographic characteristics of PSAC and caregivers and economic characteristics of households in Gamo Gofa zone, Southern Ethiopia, January, 2019.**

| Variables | Category | Cases (n = 402) | | Controls (n = 804) | |
|---|---|---|---|---|---|
| | | Frequency | % | Frequency | % |
| **Children's sex** | Male | 221 | 55.0 | 408 | 50.7 |
| | Female | 181 | 45.0 | 396 | 49.3 |
| **Children's age (years)** | ≤2 | 136 | 33.8 | 258 | 32.1 |
| | 3–5 | 266 | 66.2 | 546 | 67.9 |
| **Caregivers' age (years)** | <20 | 12 | 3.0 | 16 | 2.0 |
| | 20–29 | 133 | 33.1 | 318 | 39.5 |
| | 30–39 | 229 | 57.0 | 431 | 53.6 |
| | 40–49 | 23 | 5.7 | 35 | 4.4 |
| | ≥50 | 5 | 1.2 | 4 | 0.5 |
| **Place of residence** | Urban | 51 | 12.7 | 202 | 25.1 |
| | Rural | 351 | 87.3 | 602 | 74.9 |
| **Started education** | No | 339 | 84.3 | 646 | 80.4 |
| | Yes | 63 | 15.7 | 158 | 19.6 |
| **Number of household members** | ≤5 | 200 | 49.7 | 415 | 51.6 |
| | >5 | 202 | 50.3 | 389 | 48.4 |
| **Caregiver's occupation** | Farming | 261 | 64.9 | 477 | 59.3 |
| | Government employee | 16 | 4.0 | 54 | 6.7 |
| | Merchant | 67 | 16.7 | 166 | 20.6 |
| | Unemployed | 47 | 11.7 | 96 | 12.0 |
| | Other* | 11 | 2.7 | 11 | 1.4 |
| **Caregiver's educations status** | Can't read and write | 172 | 42.8 | 373 | 46.4 |
| | Can read and write | 84 | 20.9 | 106 | 13.2 |
| | Elementary | 100 | 24.9 | 210 | 26.1 |
| | Secondary | 39 | 9.7 | 81 | 10.1 |
| | Diploma and above | 7 | 1.7 | 34 | 4.2 |
| **Quintile of wealth index** | Highest | 82 | 20.4 | 160 | 20.0 |
| | Fourth | 70 | 17.4 | 170 | 21.1 |
| | Middle | 91 | 22.6 | 156 | 19.4 |
| | Second | 83 | 20.7 | 152 | 18.9 |
| | Lowest | 76 | 18.9 | 166 | 20.6 |

* = Daily laborer and housewife

households that had unclean latrine (AOR: 1.82, 95% CI: 1.19–2.78), and among those PSAC under caregivers who had lower mean score on KP of STH transmission (AOR = 1.85, 95% CI: 1.13–3.01) (Table 3). Details on univariable and multivariable analyses are presented in Table 3.

## Discussion

This study established context-specific evidences on determinants of STH infections among PSAC in the former Gamo Gofa Zone, Southern Ethiopia, to improve control strategies of STH.

Our study identified that WASH and behavioral related factors are significantly associated with STH infections among PSAC. Consistent with this, literatures suggest that improvements

**Table 3. Univariable and multivariable analyses of selected risk factors related to STH infection among PSAC, Gamo Gofa zone, Southern Ethiopia, January 2019.**

| Variables | Category | STH infection status | | Univariable analysis, COR (95% CI) | Multivariable analysis, AOR (95% CI) |
|---|---|---|---|---|---|
| | | Yes (n = 402) | No (n = 804) | | |
| **Place of residence** | Urban | 51 | 202 | 0.43 (0.31–0.61)** | 0.55 (0.39–0.79)* |
| | Rural | 351 | 602 | Reference | Reference |
| **Children's sex** | Male | 221 | 408 | 1.18 (0.93–1.51)** | - - |
| | Female | 181 | 396 | Reference | |
| **Children's age (years)** | ≤2 | 136 | 258 | 1.08 (0.84–1.39) | - - |
| | 3–5 | 266 | 546 | Reference | |
| **Water source** | Safe | 322 | 690 | 0.66 (0.48–0.91)** | 0.67 (0.47–0.93)* |
| | Unsafe | 80 | 114 | Reference | Reference |
| **Treat water** | No | 317 | 602 | 1.21(0.91–1.61)** | - - |
| | Yes | 85 | 197 | Reference | |
| **Distance from water source** | <30 minutes | 149 | 421 | 0.54 (0.42–0.68)** | 0.51 (0.39–0.67)* |
| | ≥30 minutes | 253 | 383 | Reference | Reference |
| **Latrine clean (n = 1157)** | No | 60 | 73 | 1.77 (1.23–2.55)** | 1.82 (1.19–2.78)* |
| | Yes | 325 | 699 | Reference | Reference |
| **Having functional hand wash facility (n = 1157)** | No | 212 | 370 | 1.33 (1.04–1.70)** | 1.36(1.04–1.77) * |
| | Yes | 173 | 402 | Reference | Reference |
| **KP score on STH prevention** | ≤5 | 385 | 751 | 1.60 (0.91–2.80)** | - - |
| | >5 | 17 | 53 | Reference | |
| **Knowledge score on STH transmission** | ≤5 | 378 | 712 | 2.04 (1.28–3.24)** | 1.85(1.13–3.01) * |
| | >5 | 24 | 92 | Reference | Reference |
| **Child hand wash habit before meal** | No | 118 | 205 | 1.21 (0.93–1.58)** | - - |
| | Yes | 284 | 599 | Reference | |
| **Child hand wash habit after defecation** | No | 208 | 373 | 1.24 (0.97–1.57)** | - - |
| | Yes | 194 | 431 | Reference | |
| **Caregivers hand wash habit after cleaning child** | No | 33 | 89 | 0.72 (0.47–1.09)** | - - |
| | Yes | 369 | 715 | Reference | |
| **Hand wash after cleaning child** | No | 33 | 89 | 0.72 (0.47–1.09)** | - - |
| | Yes | 369 | 715 | Reference | |
| **Received deworming drugs in the last year** | No | 118 | 213 | 1.15 (0.88–1.50) | - - |
| | Yes | 284 | 591 | Reference | |
| **Washing fruit or vegetables habit before eating** | No | 135 | 232 | 1.25 (0.96–1.61)** | - - |
| | Yes | 267 | 572 | Reference | |
| **Child started education** | No | 339 | 646 | 1.32 (0.95–1.81)** | - - |
| | Yes | 63 | 158 | Reference | |
| **Adequate water** | No | 108 | 240 | 0.86 (0.66–1.13) | - - |
| | Yes | 294 | 564 | Reference | |
| **Treat water** | No | 317 | 607 | 1.21 (0.91–1.61) | - - |
| | Yes | 85 | 197 | Reference | |
| **Latrine available (n = 1157)** | No | 17 | 32 | 1.06 (0.58–1.94) | - - |
| | Yes | 385 | 772 | Reference | |
| **Faeces or any dirt observed on latrine surface** | No | 270 | 566 | 0.85 (0.65–1.12)** | 0.06 (0.98–1.86) |
| | Yes | 115 | 206 | Reference | Reference |
| **Child soil eating habit** | No | 278 | 541 | 1.09 (0.84–1.41) | - - |
| | Yes | 124 | 263 | Reference | |

(*Continued*)

**Table 3.** (Continued)

| Variables | Category | STH infection status | | Univariable analysis, COR (95% CI) | Multivariable analysis, AOR (95% CI) |
| --- | --- | --- | --- | --- | --- |
| | | Yes (n = 402) | No (n = 804) | | |
| **Child shoe wear habit** | No | 184 | 354 | 1.07 (0.84–1.36) | - - |
| | Yes | 218 | 450 | Reference | |
| **Place of child body wash** | Home | 380 | 750 | 1.24 (0.75–2.07) | |
| | River | 22 | 54 | Reference | |
| **Caregiver's awareness on STH** | No | 48 | 83 | 1.18 (0.81–1.72) | - - |
| | Yes | 354 | 721 | Reference | |
| **Educational status of caregivers** | Cannot read and write | 172 | 373 | 0.45 (0.19–1.03) | - - |
| | Can read and write | 84 | 106 | 0.26 (0.11–0.61) | |
| | Elementary | 100 | 210 | 0.43 (0.18–1.01) | |
| | Secondary | 39 | 81 | 0.43 (0.17–1.05) | |
| | Diploma and above | 7 | 34 | Reference | |

**Note:** Variable (s) entered on step 1 in multivariable model were place of residence, child sex, age of caregivers, water sources, treat water, distance from water sources, latrine clean (faeces or any dirt observed on latrine surface/floor), having functional hand washing facility around latrine, wash fruits or vegetables before eating, child started education, hand wash habit after cleaning child, child hand washing habit before meal, child hand wash habit after defecation, caregiver's KP score on STH prevention, and knowledge score on STH transmission.

COR = Crude odd ratio

AOR = Adjusted odd ratio

**P ≤0.25 at univariable analysis

AOR = Adjusted odds ratio

"Reference" = comparison group

*Statistically significant at 5% level of significance in multivariable model

of WASH infrastructures and appropriate health-seeking behavior are essential for achieving sustained control and elimination of STH and many other NTDs at large [22, 23].

This study found that the odds of STH infection among PSAC who were living in urban area were lowest compared to those PSAC who were living in rural areas. This result is consistent with the findings observed in other developing countries, where STH infections were common in rural areas than urban areas [24]; the possible reason for lower odds of infection in urban area might be associated with availability of better WASH infrastructures than rural areas.

Similarly, the odds of STH infection among PSAC from households that had safe water source were lowest compared to those PSAC from households which had unsafe water source. The finding from this study corroborates with the finding of a study observed in another part of Ethiopia [19], and in other counties, such as Bangladesh [25], South Africa [26], Argentina [27] and South west China [4]. For instance, the Bangladesh study reported that "use of tube well water was associated with a 48% reduction in STH infection."

In addition, the odds of STH infection among PSAC from households with less distance from water source (<30 minutes) were lowest compared to those children from households that needed to walk longer distance (≥30 minutes) to collect water. In line with the result of this study, a systematic review and meta-analysis stated that "access to piped water was associated with lower odds of *A. lumbricoides* and *T. trichiura* infection" [23]. This is justifiable because increased access to water source can improve utilization of water for better hygiene practices which would in turn help to halt the cycle of STH transmission.

On the other hand, the odds of STH infection among PSAC children from households that had no functional hand washing facility, and those from households that had no clean latrine were highest than those who had functional hand washing facility and clean latrine. These results agree with the findings of a systematic review and meta-analysis [23, 28], and a study conducted in Uganda [29]. This finding can be possibly explained by the fact that improved sanitation and hygiene is associated with reduced odds of STH infection [15, 22, 23, 27, 28].

Likewise, our study found that the odds of STH infection among PSAC who were under caregivers with lower knowledge score on STH transmission ($\leq 5$) were nearly higher by 2 fold than those children whose caregivers had higher mean KP score. This finding supports the result of other study conducted in another part of Ethiopia [30]. The possible reason for significant association of lower knowledge on STH transmission could be related to weak social behavioral change communication intervention.

While the WHO as well as Ethiopian Ministry of Health recommend preventive chemotherapy as public health interventions to control and eliminate STH [7, 11]; in this study we have been amazed by the statistically insignificant association of deworming treatment on STH infection. This might be due to inadequate deworming coverage of PSAC and problems related to proper timing of mass drug administration (MDA), frequency of treatment and low compliance of treatment. The WHO recommends yearly deworming in communities with infection rates of 20% to 50%; however, without appropriate environmental and behavioral interventions, this may lead to re-infection rapidly after treatment. Consistent with this, a study conducted in China revealed that "statistically insignificant effect of deworming treatment on STH infection" [4].

The main strength of this study is that it identified determinants of STH with a good sample size and powered to estimate difference between groups, and it showed context-specific shortcomings of preventive chemotherapy intervention; as it will not be effective without WASH and behavioral interventions. However, in this study, the following limitations should be acknowledged. First, due to the retrospective nature of the study, data are subject to recall bias. Second, the lower positive predictive values in low-intensity settings in the Kato–Katz diagnostic technique might lead to misclassification of participants [31, 32].

## Conclusions

Given efforts required to sustain control and elimination of STH by 2030 in PSAC; this study demonstrated that the existing preventive chemotherapy should be substantially strengthened with WASH and behavioral interventions. Thus, an urgent call for action is demanding to integrate context-specific WASH interventions, particularly in rural areas. Barriers related to effective implementation of MDA for STH need to be explored in future studies.

## Supporting information

**S1 Questionnaire. Data collection tool for face-to-face interview.**
(DOCX)

**S1 Dataset. SPSS dataset.**
(SAV)

## Acknowledgments

Authors would like thank the study participants, data collectors, supervisors, zonal health office heads, district health office heads and NTDs focal points in all selected districts of the study area for their continued support during implementation of the study.

## Author Contributions

**Conceptualization:** Mekuria Asnakew Asfaw.

**Data curation:** Mekuria Asnakew Asfaw.

**Formal analysis:** Mekuria Asnakew Asfaw.

**Funding acquisition:** Mekuria Asnakew Asfaw, Teklu Wegayehu, Tigist Gezmu.

**Investigation:** Mekuria Asnakew Asfaw, Teklu Wegayehu, Tigist Gezmu, Alemayehu Bekele, Zeleke Hailemariam, Teshome Gebre.

**Methodology:** Mekuria Asnakew Asfaw, Teklu Wegayehu, Tigist Gezmu, Alemayehu Bekele, Zeleke Hailemariam, Teshome Gebre.

**Project administration:** Mekuria Asnakew Asfaw, Teklu Wegayehu, Tigist Gezmu.

**Resources:** Mekuria Asnakew Asfaw.

**Software:** Mekuria Asnakew Asfaw.

**Supervision:** Mekuria Asnakew Asfaw, Teklu Wegayehu, Tigist Gezmu, Zeleke Hailemariam, Teshome Gebre.

**Validation:** Mekuria Asnakew Asfaw.

**Visualization:** Mekuria Asnakew Asfaw, Teklu Wegayehu, Tigist Gezmu, Teshome Gebre.

**Writing – original draft:** Mekuria Asnakew Asfaw.

**Writing – review & editing:** Mekuria Asnakew Asfaw, Teklu Wegayehu, Tigist Gezmu, Alemayehu Bekele, Zeleke Hailemariam, Teshome Gebre.

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
