## [Decision Letter · Decision Letter 0]

28 Jul 2020

PONE-D-20-19992

Determinants of soil-transmitted helminth infection among pre-school-age children in Gamo Gofa zone, Southern Ethiopia: a case-control study

PLOS ONE

Dear Dr. Asfaw,

Thank you for submitting your manuscript to PLOS ONE. After careful consideration, we feel that it has merit but does not fully meet PLOS ONE’s publication criteria as it currently stands. Therefore, we invite you to submit a revised version of the manuscript that addresses the points raised during the review process.

Your manuscript has been revised by two reviewers and they have raised important points you have to respond to.

We look forward to receiving your revised manuscript.

Kind regards,

Kebede Deribe, BSc, MPH, PhD

Academic Editor

PLOS ONE

Journal Requirements:

3. We note you have included a table to which you do not refer in the text of your manuscript. Please ensure that you refer to Table 4 in your text; if accepted, production will need this reference to link the reader to the Table.

Reviewers' comments:

Reviewer's Responses to Questions

**Comments to the Author**

1. Is the manuscript technically sound, and do the data support the conclusions?

Reviewer #1: Yes

Reviewer #2: Yes

2. Has the statistical analysis been performed appropriately and rigorously? 

Reviewer #1: Yes

Reviewer #2: No

3. Have the authors made all data underlying the findings in their manuscript fully available?

Reviewer #1: Yes

Reviewer #2: Yes

4. Is the manuscript presented in an intelligible fashion and written in standard English?

Reviewer #1: Yes

Reviewer #2: Yes

5. Review Comments to the Author

Reviewer #1: Determinants of soil-transmitted helminth infection among pre-school-age children in Gamo Gofa zone, Southern Ethiopia: a case-control study

The authors present the findings on determinants of STH infection among PSAC in a study area in Southern Ethiopia. The study provides evidence on the association of STH and WASH factors among PSAC group of children. Therefore, this is a relevant study and of interest to STH control programs.

General comments:

The entire manuscript should be checked and proofread for grammatical and spelling corrections

Specific comments:

Abstract:

Include the cut-off for lower mean score for KP, line 47.

Introduction:

Generally well written.

Methods:

Avoid a one sentence paragraphs. Please provide more and explicit information on the study design. Lines 96-97.

The stated age group of 1-5 years is not the generally used PSAC age group, for comparability of the results, amend the analysis using the generally used PSAC age group.

State how STH infection was defined as an outcome variable, was it any STH positive result? Line 119.

How was the questionnaire administered? Paper or technology-based? State how the method used guaranteed data quality and integrity. Lines 124-126.

How were scores for KP created? Lines 145-146

Usually, multivariable logistic regression model is build following a univariable logistic regression and variable selection process. Was univariable logistic regression conducted? Perhaps a table summarizing this result is necessary. However, I have noted that a bivariate analysis was conducted using chi-square test, you don’t need to present its results. Just note that univariable logistic regression is different with chi-square test. Why was p<0.25 considered for the bivariable analysis?

State the ethical approval number if the study was reviewed and approved. State if assent was obtained from children. Why was written consent not obtained from the parents/household heads? Lines 152-155

Results:

Give a paragraph summarizing the infection prevalence by each individual species and any STTH among the participants.

Instead of running the logistic regression model on only any STH, considering doing this analysis on each STH species as well. This would inform on the WASH variation on individual species.

State in the methods how latrine cleanliness was defined, lines 176.

Tables 3 and 4: Instead of writing “1” write “reference”, since this refers to the reference category. Put table note explaining what “--” means. Separate the multivariable analysis on its own table. Delete chi-square test results and put the univariable logistic regression results instead. Give the definition of unsafe/safe water source. Where are the results of multivariable analysis in table 4?

Discussion:

Generally written well but should be improved after addressing the above concerns touching on methods and results.

Reviewer #2: Review Report for Determinants of soil-transmitted helminth infection among pre-school-age children in Gamo Gofa zone, Southern Ethiopia: a case-control study

Thank you for the opportunity to review this paper that explores the role of WASH in STH infection among pre-school-aged children. While the study provides new information, there is need for more detail on how the study was conducted. The tables also need to organized for provide sharper message. For example, table one wold look at child level factor based on information from primary care givers. Next can look at wash specific factors. The discussion should also be consolidated to comparison of the current study with other published studies

A few specific questions

1. Are there any contextual differences in the kebeles selected?

2. There is no justification provided why PSAC not SAC, the references provided in the first paragraph on the introduction need to be updated.

3. This statement is true for SAC not sure about PSAC

“It has been provided for all PSAC for many years in endemic 70 countries including Ethiopia in areas where the baseline prevalence of any soil-transmitted 71 infections is 20% or higher among children, in order to control and eliminate STH”

4. Description of Kato katz procedure is not described.

5. Ethics statement mentions oral consent, I appreciate different settings have varied ethical requirements, just wondering whether in this setting oral consent is sufficient

6. How was the sampling done

6. PLOS authors have the option to publish the peer review history of their article (what does this mean?). If published, this will include your full peer review and any attached files.

Reviewer #1: No

Reviewer #2: No

---

## [Author Response · Author response to Decision Letter 0]

14 Sep 2020

Author Response

In general, authors highly valued the reviewers’ and editor’s comments. Accordingly, amendments have been made.

Response to editor’s comments

Journal Requirements:

Response: The manuscript has been modified to satisfy all the journal requirements.

2. Please include additional information regarding the survey or questionnaire used in the study and ensure that you have provided sufficient details that others could replicate the analyses.

Response: The survey questionnaire has been included as additional information in the revised version. 

3. We note you have included a table to which you do not refer in the text of your manuscript. Please ensure that you refer to Table 4 in your text; if accepted, production will need this reference to link the reader to the Table.

Response: We appreciate the editor’s comments, and now correction has been made, line 220.

4. After careful consideration, we feel that it has merit but does not fully meet PLOS ONE’s publication criteria as it currently stands. Therefore, we invite you to submit a revised version of the manuscript that addresses the points raised during the review process.

Response: Authors would like to thank and acknowledge the editors’ concern. Accordingly, the manuscript has been amended to meet PLOS ONE’s publication criteria.

Response to Reviewers' comments

Reviewer #1

1. The entire manuscript should be checked and proofread for grammatical and spelling corrections.

Response: The manuscript has been proofread with editing made wherever necessary.

2. Abstract: Include the cut-off for lower mean score for KP, line 47

Response: Authors appreciate the reviewer’s concern. Agreed, correction has been made, line 48. 

3. Methods: Avoid a one sentence paragraphs. Please provide more and explicit information on the study design. Lines 96-97

Response: Agreed, now amendments have been made in the revised version, lines 101-104. 

4. Methods: The stated age group of 1-5 years is not the generally used PSAC age group, for comparability of the results, amend the analysis using the generally used PSAC age group.

Response: Authors appreciated for the reviewer’s concern. However, since there is no consistent definition of PSAC in the existing literatures, in this study, all children aged 1 to 5 years who are not yet attending (primary) school were considered as pre-school children (lines 108-110) , as supported by WHO (please see at: http://whqlibdoc.who.int/publications/2006/9241547103_eng.pdf) and other studies, such as https://doi.org/10.1186/s12879-018-3289-0 and https://doi.org/10.1186/s40249-019-0561-5. Moreover, in Ethiopia, almost all children start primary school after the age of 5. 

5. Methods: State how STH infection was defined as an outcome variable, was it any STH positive result? Line 119

Response: Agreed. In our study, the STH infection status was defined as being positive or negative for any STH species. Lines 128-129

6. How was the questionnaire administered? Paper or technology-based? 

Response: Paper based questionnaire was administered through face-to-face interviews.

7. State how the method used guaranteed data quality and integrity. Lines 124-126

Response: Authors appreciated the concern raised by reviewer; however, we think that the issues raised by the reviewer were addressed at quality control section of the manuscript (See at lines 159-163 in the revised version).

8. Methods: How were scores for KP created? Lines 145-146

Response: We appreciated for the question. We measured knowledge score on STH transmission using 11 questions, and score on knowledge and practice on STH prevention was computed using 12 questions by counting value within a case in SPSS version 25.0. Lines 171-173 in the revised version

9. Methods: Usually, multivariable logistic regression model is built following a univariable logistic regression and variable selection process. Was univariable logistic regression conducted? Perhaps a table summarizing this result is necessary. However, I have noted that a bivariate analysis was conducted using chi-square test; you don’t need to present its results. Just note that univariable logistic regression is different with chi-square test. Why was p<0.25 considered for the bivariable analysis? 

Response: Authors are grateful for the reviewer’s comments. Note that data presented as ‘bivariate analysis’ should be considered as ‘univariable logistic regression’. Again, note that p < 0.25 was not used for the ‘bivariate analysis; we did it for ‘univariable logistic regression’ to enter candidate variables into ‘multivariable logistic regression’. The reason for using p < 0.25 was to improve the chances of remaining potential variables in the multivariable model. With consideration given to reviewer’s comments, now amendment has been made in the revised version. Lines 201-212

10. Methods: State the ethical approval number if the study was reviewed and approved. State if assent was obtained from children. Why was written consent not obtained from the parents/household heads? Lines 152-155

Response: Thank you for the comments. Now the ethical issue is addressed. The reference number was: CMHS/11222/111). Assent was not obtained from PSAC since we believe that caregivers are responsible on behalf of them, lines 183-185.

11. Results: Give a paragraph summarizing the infection prevalence by each individual species and any STTH among the participants.

Response: With regard to infection prevalence by each individual STH species, overall, ascariasis was the most prevalent (27.7%), followed by trichiurasis (11.9%) and hookworms (4.6%), and 8% PSAC were infected with two STH species (ascariasis and trichiurasis), lines 190-193.

12. Results: Instead of running the logistic regression model on only any STH, considering doing this analysis on each STH species as well. This would inform on the WASH variation on individual species.

Response: Authors appreciated for the reviewer’s comments; however, from the very beginning of the study, the sample size was calculated based on infection status of any STH species as (cases vs. controls) by taking our hypothesis - any STH infection have shared WASH factors into consideration. 

13. Results: State in the methods how latrine cleanliness was defined, lines 176.

Response: Authors thanks for reviewer comments. The issue has been addressed. Lines 174-175

14. Results: Tables 3 and 4: Instead of writing “1” write “reference”, since this refers to the reference category. Put table note explaining what “--” means. Separate the multivariable analysis on its own table. Delete chi-square test results and put the univariable logistic regression results instead. Give the definition of unsafe/safe water source. Where are the results of multivariable analysis in table 4?

Response: Agreed. The issue has been addressed in the revised version. In the table 3 of the revised manuscript, the multi-variable analysis was done for variables with P<0.25 in univariable analysis. Please note that we used backward stepwise method to enter potential variables into the multi-variable analysis, which result in seven variables to be remained in the final model, as it is indicated with an AOR (95% CI) in the table 3. In addition, note that the multivariable model was placed at side of univariable analysis just for comparison purpose, if it does not make sense in this way it is possible to separate simply. Table 4 is edited as table 3. Lines 231-235

15. Discussion: Generally written well but should be improved after addressing the above concerns touching on methods and results.

Response: Thank you. Now some improvements have been made in the revised version.

Reviewer #2

1. While the study provides new information, there is need for more detail on how the study was conducted. The tables also need to organized for provide sharper message. For example, table one wold look at child level factor based on information from primary care givers. Next can look at wash specific factors. The discussion should also be consolidated to comparison of the current study with other published studies

Response: Agreed. Amendments have been made in the revised version. 

2. Are there any contextual differences in the kebeles selected?

Response: We appreciated for the reviewer’s question. We assumed that there are socio-demographic and WASH related risk factors variations across the selected kebeles that that could contribute for difference in prevalence of STH.

3. There is no justification provided why PSAC not SAC, the references provided in the first paragraph on the introduction need to be updated.

Response: Authors valued the reviewer’s comment. Agreed., revision has been made as: Since there is no consistent definition of PSAC in the existing literatures, in this study, all children aged 1 to 5 years who are not yet attending (primary) were considered as pre-school children (lines 108-110) , as supported by WHO guideline (see at: http://whqlibdoc.who.int/publications/2006/9241547103_eng.pdf. 

In fact, in Ethiopia, almost all children start primary school after the age of 5. In addition, for comparison purpose, other studies, which were conducted in South Africa and Uganda also considered PSAC as children aged 1 to 5. 

4. This statement is true for SAC not sure about PSAC

“It has been provided for all PSAC for many years in endemic 70 countries including Ethiopia in areas where the baseline prevalence of any soil-transmitted 71 infections is 20% or higher among children, in order to control and eliminate STH”

Response: We think as it may works for PSAC. But we found the statement -“between 2001 and 2009, the number of school-age children benefiting from deworming programmes had tripled to more than 200 million in over 60 countries.” (Weekly Epidemiological Record, 2011, 25(86):257–268). 

5. Description of Kato katz procedure is not described.

Response: Now the Kato katz procedure has been described in the revised version, lines 144-157.

6. Ethics statement mentions oral consent, I appreciate different settings have varied ethical requirements, just wondering whether in this setting oral consent is sufficient

Response: Oral and written consent was received from district administrators and heads of the households before the data collection began. Lines 183-184 

7. How was the sampling done?

Response: We thank for the important question. The sampling was done by taking probability proportionate to sample size into account based on the number of cases in the cross-sectional study; which was conducted ahead of this study in the same study area, lines 122-126

---

## [Decision Letter · Decision Letter 1]

21 Oct 2020

PONE-D-20-19992R1

Determinants of soil-transmitted helminth infections among pre-school-age children in Gamo Gofa zone, Southern Ethiopia: a case-control study

PLOS ONE

Dear Dr. Asfaw,

Thank you for submitting your manuscript to PLOS ONE. After careful consideration, we feel that it has merit but does not fully meet PLOS ONE’s publication criteria as it currently stands. Therefore, we invite you to submit a revised version of the manuscript that addresses the points raised during the review process.

We look forward to receiving your revised manuscript.

Kind regards,

Kebede Deribe, BSc, MPH, PhD

Academic Editor

PLOS ONE

Reviewers' comments:

Reviewer's Responses to Questions

**Comments to the Author**

1. If the authors have adequately addressed your comments raised in a previous round of review and you feel that this manuscript is now acceptable for publication, you may indicate that here to bypass the “Comments to the Author” section, enter your conflict of interest statement in the “Confidential to Editor” section, and submit your "Accept" recommendation.

Reviewer #1: (No Response)

2. Is the manuscript technically sound, and do the data support the conclusions?

Reviewer #1: Yes

3. Has the statistical analysis been performed appropriately and rigorously? 

Reviewer #1: Yes

4. Have the authors made all data underlying the findings in their manuscript fully available?

Reviewer #1: Yes

5. Is the manuscript presented in an intelligible fashion and written in standard English?

Reviewer #1: Yes

6. Review Comments to the Author

Reviewer #1: The authors have done a great job to address explicitly all my earlier comments. However, as I had stated earlier, some grammatical and sentence construction errors remain that need to be addressed. Perhaps, an independent copy editing need to be considered. Additionally, the authors need to make it clear whether they used P<0.25 (line 202) or P≤0.25 (line 209) for univariable analysis, make this consistent throughout the text. In table 3, indicate the word “Reference” on the comparison categories.

7. PLOS authors have the option to publish the peer review history of their article (what does this mean?). If published, this will include your full peer review and any attached files.

Reviewer #1: No

---

## [Author Response · Author response to Decision Letter 1]

2 Nov 2020

Authors’ Responses

Authors highly valued and appreciate the editor’s and reviewers’ comments. The manuscript has been modified accordingly.

Response to Reviewer’s comments

Reviewer #1

1. The authors have done a great job to address explicitly all my earlier comments. However, as I had stated earlier, some grammatical and sentence construction errors remain that need to be addressed. Perhaps, an independent copy editing need to be considered. 

Response: Authors are grateful for the reviewer’s comments. With special attention given to grammatical and sentence construction errors, the entire manuscript has been checked and proofread for grammatical and spelling corrections. 

2. Additionally, the authors need to make it clear whether they used P<0.25 (line 202) or P≤0.25 (line 209) for univariable analysis, make this consistent throughout the text.

Response: Agreed, thank you. Correction has been made. And please note that we used at P ≤0.25 to select potential variables for fitting into multi-varable model. We made correction on the revised version of the manuscript accordingly.

3. In Table 3, indicate the word “Reference” on the comparison categories.

Response: Agreed, correction has been made in the revised version.

---

## [Editor Report · Decision Letter 2]

6 Nov 2020

PONE-D-20-19992R2

Determinants of soil-transmitted helminth infections among pre-school-aged children in Gamo Gofa zone, Southern Ethiopia: a case-control study

PLOS ONE

Dear Dr. Asfaw,

Thank you for submitting your manuscript to PLOS ONE. After careful consideration, we feel that it has merit but does not fully meet PLOS ONE’s publication criteria as it currently stands. Therefore, we invite you to submit a revised version of the manuscript that addresses the points raised during the review process.

We look forward to receiving your revised manuscript.

Kind regards,

Kebede Deribe, BSc, MPH, PhD

Academic Editor

PLOS ONE

Additional Editor Comments (if provided):

Table 3: Faeces or any dirt observed on latrine surface, in the Multi-variable analysis, AOR (95%CI) column provide the reference category.

Table 3. Educational status of caregivers, provide the Univariable analysis, COR (95%CI) for the categories

• Can read and write

• Elementary

• Secondary

---

## [Author Response · Author response to Decision Letter 2]

9 Nov 2020

Authors’ Responses

Authors are grateful for the editor’s comments, and the manuscript has been amended accordingly.

Response to editors’ comments

1. Table 3: Faeces or any dirt observed on latrine surface, in the Multi-variable analysis, AOR (95%CI) column provide the reference category.

 Response: Agreed, correction has been made in the table at line 235.

2. Table 3. Educational status of caregivers, provide the Univariable analysis, COR (95%CI) for the categories 

• Can read and write

• Elementary

• Secondary

 Response: Agreed, correction has been made in the table at line 235.

---

## [Editor Report · Decision Letter 3]

27 Nov 2020

Determinants of soil-transmitted helminth infections among pre-school-aged children in Gamo Gofa zone, Southern Ethiopia: a case-control study

PONE-D-20-19992R3

Dear Dr. Asfaw,

We’re pleased to inform you that your manuscript has been judged scientifically suitable for publication and will be formally accepted for publication once it meets all outstanding technical requirements.

Kind regards,

Kebede Deribe, BSc, MPH, PhD

Academic Editor

PLOS ONE

---

## [Editor Report · Acceptance letter]

2 Dec 2020

PONE-D-20-19992R3 

Determinants of soil-transmitted helminth infections among pre-school-aged children in Gamo Gofa zone, Southern Ethiopia: a case-control study 

Dear Dr. Asfaw:

I'm pleased to inform you that your manuscript has been deemed suitable for publication in PLOS ONE. Congratulations! Your manuscript is now with our production department. 

Kind regards, 

on behalf of

Dr. Kebede Deribe 

Academic Editor

PLOS ONE